# TopoImb: Toward Topology-level Imbalance in Learning from Graphs

**Tianxiang Zhao[†], Dongsheng Luo[‡], Xiang Zhang[†], Suhang Wang[†]**
[†]The Pennsylvania State University, University Park, PA
[‡] Florida International University, Miami, FL
{tkz5084, xzz89, szw494}@psu.edu, dluo@fiu.edu

## Abstract

Graph serves as a powerful tool for modeling data that has an underlying structure in non-Euclidean space, by encoding relations as edges and entities as nodes. Despite developments in learning from graph-structured data over the years, one obstacle persists: graph imbalance. Although several attempts have been made to target this problem, they are limited to considering only class-level imbalance. We argue that for graphs, the imbalance is likely to exist at the sub-class level in the form of infrequent topological motifs. Due to the flexibility of topology structures, graphs could be highly diverse, and learning a generalizable classification boundary would be difficult. Therefore, several majority topology groups may dominate the learning process, rendering others under-represented. To address this problem, we propose a new framework TopoImb and design (1) a topology extractor, which automatically identifies the topology group for each instance with explicit memory cells, (2) a training modulator, which modulates the learning process of the target GNN model to prevent the case of topology-group-wise under-representation. TopoImb can be used as a key component in GNN models to improve their performances under the data imbalance setting. Analyses on both topology-level imbalance and the proposed TopoImb are provided theoretically, and we empirically verify its effectiveness with both node-level and graph-level classification as the target tasks.

## 1  Introduction

Graphs are ubiquitous in the real world [1, 2], such as social networks, finance networks and brain networks. Hence, graph-based learning is receiving increasing attention due to its advantage in modeling relations/interactions (as edges) of entities (as nodes). Recently, graph neural networks (GNNs) have shown great ability in representation learning on graphs, facilitating various domains [3, 4]. However, similar to other domains like computer vision and natural language processing, learning on graphs could also suffer from the problem of data imbalance [5, 6]. Imbalanced sizes of labeled data would harm the classifier, and render the classification boundary dominated by majority groups. Numerous efforts have been made addressing this problem [7–9]. Typically in these works, data imbalance is discussed at the class level: instances of some classes, i.e., majority classes, are much larger in quantity than other classes, i.e., minority classes. For example in imbalanced graph learning strategies, GraphSMOTE [10] addresses node imbalance by inserting new nodes of the minority classes into the given graph. ReNode [11] considers coverage of propagated influence with labeled nodes utilizing re-weighting.

Although these methods help improve the performance of graph learning algorithms on imbalanced training samples, the deficiencies of constraining data imbalance to mere class level are evident. Due to the diversity of inputs, data imbalance in graphs could impair the classifier at the sub-class level. Not only minority classes, but some minority topology groups inside each class can also be under-represented. As a result, graph learning models may get stuck in a local optimum by over-fitting to those majority groups and fail to learn effectively on those minority groups. We call this sub-class level imbalance in graphs as **Topology-level Imbalance** problem. An example is provided in Fig. 1.

T. Zhao et al., TopoImb: Toward Topology-level Imbalance in Learning from Graphs. *Proceedings of the First Learning on Graphs Conference (LoG 2022)*, PMLR 198, Virtual Event, December 9–12, 2022.

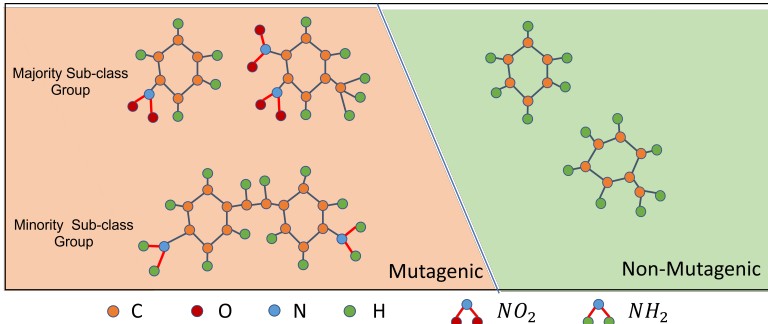

**Figure 1:** Example of sub-class topology level graph imbalance on dataset Mutag [12]. Molecular graphs of the Mutagenic class have two topology groups, one with motif $NO_2$ and another one with motif $NH_2$ [13]. The $NO_2$ group is much larger in quantity compared to the $NH_2$ group.

in a molecular dataset, a property (label) could be caused by multiple atom motifs, and rationales relating to infrequent motifs would be difficult to capture due to this imbalance problem.

In contrast to the class-level imbalance problem, where the label distribution is (partially) explicitly given training nodes/graphs, this sub-class *Topology Imbalance* is more general, but at the same time much more challenging. First, the distribution of topology groups is implicit. Existing techniques, including *cost-sensitive learning* [8, 11] and *re-sampling* [10, 14], manually set a weighting function based on certain assumptions on training data. They inevitably involve hyper-parameters to be manually preset or tuned by cross-validation, which prevents them from directly handling the Topology Imbalance problem. Second, features of topology groups are rooted in both node attributes and edge existences. Difficulties in measuring the similarity of topology complicate the discovery of sub-class-level groups and the modulation process. Third, the number of topology structures grows exponentially with the graph size, hence the discovery of poorly-learned topology groups should be learned efficiently from model performance, and need to adapt to evolving model behaviors.

To tackle the aforementioned issues, in this work, we make the first attempt to address graph imbalance at a more fine-grained level, i.e., sub-class level, by considering underrepresented topology groups. Concretely, we develop a novel framework TopoImb, to augment the training process with an imbalance-sensitive modulation mechanism. TopoImb adopts a topology extractor to explicitly model and dynamically update the discovery of topology groups. Based on that, a training modulator automatically and adaptively modulates the training process by assigning importance weights to training instances that are under-represented.

Altogether, this work makes the following three-fold contributions: (1) To our best knowledge, we are the first to analyze the sub-class level imbalance problem in graph learning. (2) We propose a plug-and-play framework as a general solution to the limitations in existing graph learning methods under topology imbalanced settings, and provide a theoretical analysis on it. (3) We adopt both synthetic and real-world datasets to reveal the failure modes of existing deep graph learning methods and verify the effectiveness of TopoImb in improving the generalization of various graph neural networks.

## 2   Preliminary

### 2.1   Problem of Sub-class Level Imbalance

One critical obstacle in addressing sub-class level imbalance lies in identifying sub-class groups. In many real-world scenarios, there could be multiple different prototypes for each class, as shown in Fig. 1. This phenomenon can be described in the following Gaussian Mixture distribution:

$$p(\mathbf{x} \mid c) = \sum_{k=1}^{K} \pi_k^c \mathcal{N}(\mathbf{x} \mid \boldsymbol{\mu}_k^c, \boldsymbol{\rho}_k^c), \tag{1}$$

where $\mathbf{x}$ is the node (graph) embedding, and $p(\mathbf{x} \mid c)$ shows that there are $K$ sub-class distribution centers (prototypes/templates) for instances belonging to class $c$. Each topology group in class $c$ is

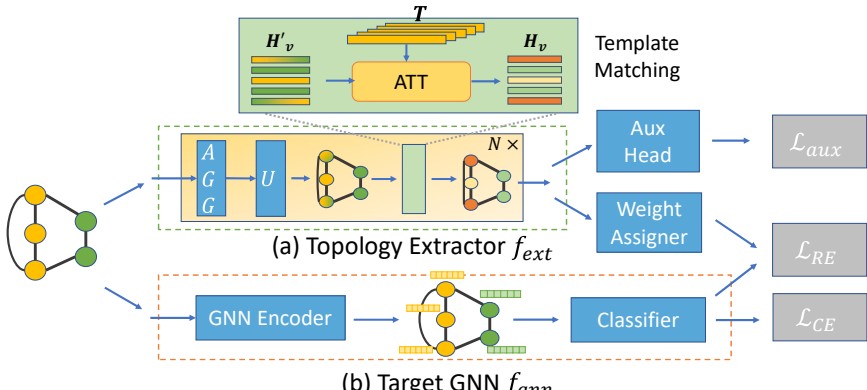

**Figure 2:** Overview of TopoImb framework. Its main components include (a) a topology extractor $f_{ext}$, and (b) the target GNN $f_{gnn}$. A weight assigner is implemented on top of the topology extractor to modulate learning of $f_{gnn}$, and an auxiliary task head is adopted to guide topology extraction.

modeled using a multivariate Gaussian distribution with a mean vector $\boldsymbol{\mu}_k^c$ and a covariance matrix $\boldsymbol{\rho}_k^c$. $\pi_k$ denotes the mixing coefficients which meets the condition: $\sum_{k=1}^K \pi_k^c = 1$. In the ideal balanced setting, each topology group should be the same in quantity: $\forall k, \pi_k^c = 1/K$. However, in most real-world cases those topology groups would follow a long-tail distribution, and those minority groups would have much smaller mixing coefficients compared to majority topology groups. This imbalance problem adds difficulty to the learning process. With training signals dominated by majority instances, the trained model could underfit and perform poorly on those minority groups in class $c$. However, unlike class-level imbalance, there is no explicit supervision of sub-class groups and it is difficult to obtain prior knowledge on imbalance ratios, making this problem challenging.

## 2.2   Notations

In this work, we conduct analysis on both the node-level task, *semi-supervised node classification*, and graph-level task, *graph classification*. We use $\mathcal{G} = \{\mathbb{V}, \mathcal{E}; \boldsymbol{F}, \boldsymbol{A}\}$ to denote a graph, where $\mathbb{V} = \{v_1, \dots, v_n\}$ is a set of $n$ nodes and $\mathcal{E} \in \mathbb{V} \times \mathbb{V}$ is the set of edges. Nodes are represented by the attribute matrix $\boldsymbol{F} \in \mathbb{R}^{n \times d}$ ($d$ is the node attribute dimension). $\mathcal{E}$ is described by the adjacency matrix $\boldsymbol{A} \in \mathbb{R}^{n \times n}$. If node $v_i$ and $v_j$ are connected then $A_{i,j} = 1$; otherwise $A_{i,j} = 0$.

For *node classification*, each node $v_i$ is accompanied by a label $Y_i \in \mathbb{C}$, where $\mathbb{C}$ is a set of labels. $\mathbb{V}^L \subset \mathbb{V}$ is the labeled node set and usually we have $|\mathbb{V}^L| \ll |\mathbb{V}|$. The objective is to train a GNN $f$ that maps each node to its class. For *graph classification*, a set of graphs $\mathbb{G} = \{\mathcal{G}_1, \dots, \mathcal{G}_m\}$ is available and each graph $\mathcal{G}_i$ has a label $Y_i \in \mathbb{C}$. Similarly, a labeled set $\mathbb{G}^L \subset \mathbb{G}$ is used to train a GNN model $f$, which maps a graph to one of the $C$ classes, i.e., $f : \{\boldsymbol{F}, \boldsymbol{A}\} \mapsto \{1, 2, \dots, C\}$.

## 3   Methodology

In this section, we introduce TopoImb, a plug-and-play framework to address the sub-class level imbalance problem during the training of GNNs. Inspired by analysis in Section 2.1, we first design a topology extractor to identify under-represented topology groups and then modulate training of the target GNN in an imbalance-aware manner. The framework is summarized in Fig. 2.

### 3.1   Topology Group Discovery

In graphs, due to the difficulties in modeling node attributes and edge distributions, it is challenging to identify latent sub-class topology groups. To successfully obtain meaningful topology groups, the topology extractor needs to satisfy the following requirements:

- Simultaneous modeling of both node attributes and edges, which is the basic requirement for modeling topology distributions as they are intertwined in latent graph generation process [1, 2].
- Able to learn with high data efficiency. The size of some topology groups is small, therefore the grouping of similar structures should be encouraged intrinsically in design.

- Representation power should be on par with 1-Weisfeiler-Lehman (WL) test, which is the upper bound of representation power for most GNNs [15]. It needs to be guaranteed that topology structures distinguishable by GNNs can also be distinguished by the extractor.

As shown in [16], most GNNs can be summarized in the message-passing framework. Let $\mathbf{h}_{v_i}^l$ be the embedding of node $v_i$ in $l$-th layer of GNN. Then the embedding of $v_i$ in the $(l+1)$-th layer is updated with aggregated messages from its neighbors $\mathcal{N}(v_i)$ as: $\boldsymbol{h}_{v_i}^{l+1} = \mathrm{U}_l(\boldsymbol{h}_{v_i}^l, \mathrm{AGG}(\{\mathrm{M}_l(\boldsymbol{h}_{v_i}^l, \boldsymbol{h}_{v_j}^l, A_{i,j}); v_j \in \mathcal{N}(v_i)\}))$, where $\mathrm{M}_l, \mathrm{AGG}, \mathrm{U}_l$ are the message creation, aggregation, and node update functions, respectively. Based on previously discussed requirements, we design our extraction module $f_{ext}$ parameterized by $\phi$ as follows: (1) AGG() is implemented as a summation of messages. The aggregation of neighborhood information should guarantee that embeddings of nodes with different neighborhood structures remain distinct and distinguishable. As shown in [15], summation of a set can guarantee the injection property and preserves the representation power to be the same as 1-WL algorithm. (2) Inspired by [17], we design a topology extraction module with external memory cells to store templates of node embeddings at each layer. Each memory cell represents a template and encourages intrinsic topology-wise grouping by mapping similar nodes to close embeddings. The templates will encode distinct structural semantics and increase the data efficiency. As shown in [18], explicit memory can greatly improve learning in a weakly-supervised scenario. An overview of the model is provided in Fig. 2(a).

Specifically, let $\boldsymbol{T}^{l+1} = [\boldsymbol{t}_1^{l+1}, \ldots, \boldsymbol{t}_{K+1}^{l+1}]$ be a matrix consisting of $K+1$ templates for the $(l+1)$-th layer, where $\boldsymbol{t}_k^{l+1}$ is the $k$-th topology template. We use the first $K$ templates to capture informative structures and the last one as the default to encode outliers and uninformative structures that might exist in the graph. To help learn informative representations of each node, we will first match node with template at $l$-th layer as:

$$\boldsymbol{z}_{v_i}^{l+1} = \mathrm{MLP}(\boldsymbol{h}_{v_i}^l, \sum_{v_j \in \mathcal{N}(v_i)} \boldsymbol{h}_{v_j}^l), \quad S_{v_i,k}^{l+1} = \begin{cases} \mathrm{Attention}(\boldsymbol{z}_{v_i}^{l+1}, \boldsymbol{t}_k^{l+1}), & k \in \{1, 2, ..., K\} \\ \delta, & k = K+1 \end{cases} \tag{2}$$

where $S_{v_i,k}^{l+1}$ measures similarity between the $(l+1)$-hop ego-graph centered at $v_i$ and template $\boldsymbol{t}_k^{l+1}$. In practice, attention is implemented as an inner product. Other forms of attention networks, such as a single-layer feedforward neural network [19] can also be adopted. Then, the representation of each node would be updated by mapping to these templates as:

$$\boldsymbol{h}_{v_i}^{l+1} = \sum_{k=1}^{K} \mathrm{softmax}(S_{v_i,k}^{l+1}) \cdot \boldsymbol{t}_k^{l+1} \tag{3}$$

This design encourages a structured embedding distribution. After training, this module will be able to automatically discover topology groups utilizing template selection. Nodes with similar local topologies will be mapped to similar regions in the high-dimensional embedding space.

## 3.2 GNN Training Modulation

The original cross-entropy learning objective for GNN $f_{gnn}$ is as follows:

$$\min_{\theta} \mathcal{L}_{CE} = -\sum_{u \in \mathbb{L}} \sum_c \mathbf{1}(Y_u == c) \cdot \log(P_u[c]), \tag{4}$$

where $\theta$ is the set of module parameters and $P_u[c]$ is the probability of instance $u$ belonging to class $c$ predicted by $f_{gnn}$. For node classification task, $\mathbf{P}_u = f_{gnn}(\boldsymbol{F}, \boldsymbol{A}; v_u)$ and $\mathbb{L} \triangleq \mathbb{V}^L$. For graph classification task, $\mathbf{P}_u = f_{gnn}(\mathcal{G}_u)$ and $\mathbb{L} \triangleq \mathbb{G}^L$. However, cross-entropy can be easily dominated by simple majority instances [20]. As a result, the trained model may perform poorly on examples with minority topology structures, which would be easily misclassified.

With the obtained topology extractor, we propose to modulate the training of GNNs to emphasize those minority groups and update it on the graph distribution regions which are not learned well. Different modulation mechanisms can be designed, and we adhere to instance re-weighting in this work to keep the simplicity, leaving the exploring of other modulation mechanisms as future work.

Concretely, we concatenate node hidden representations $\{\boldsymbol{H}^l, l = 1, 2, ...\}$ obtained from topology extractor as $\boldsymbol{H} \in \mathbb{R}^{|\mathbb{V}| \times D}$, and use a 2-layer MLP $g_{wt}$ parameterized by $\varphi$ to predict weights. $D$

is the dimension of concatenated node embeddings. For graph classification task, we further adopt global pooling to get graph-level embedding $\boldsymbol{H}_{\mathcal{G}} \in \mathbb{R}^{1 \times D}$. In this work, we use attention-based pooling to get graph-level embeddings. Then, we predict the modulation weight for $v_i$ as:

$$w_i' = g_{wt}(\boldsymbol{H}_i), \quad w_i = \frac{w_i'}{\sum_{i=1}^{b} w_i'} \cdot b \tag{5}$$

where $\boldsymbol{H}_i$ is the embedding of node $v_i$ for node classification and embedding of graph $\mathcal{G}_i$ for graph classification. $b$ denotes the batch size and is also used to normalize predicted weights. This normalization guarantees that the summation of $w$ would remain as 1, and enables competition in the assignment of weights. With the obtained $w$, TopoImb modulates training of the target GNN as:

$$\min_{\theta} \mathcal{L}_{RE} = -\sum_{u \in \mathbb{L}} \sum_{c} w_i \cdot \mathbf{1}(Y_u == c) \cdot \log(P_u[c]). \tag{6}$$

The modulation performance would also be utilized to guide the discovery of topology groups. Inspired by [21], we develop a min-max reward signal for learning this modulation from an adversarial training perspective:

$$\min_{\theta} \max_{\phi, \varphi, \mathbb{T}} \mathcal{L}_{RE} = -\sum_{u \in \mathbb{L}} \sum_{c} w_i \cdot \mathbf{1}(Y_u == c) \cdot \log(P_u[c]). \tag{7}$$

$\mathbb{T} = \{\boldsymbol{T}^1, \boldsymbol{T}^2, ...\}$ is the set of all templates. This adversarial objective would encourage $f_{ext}$ to identify topology groups where $f_{gnn}$ is poorly learned and assign them with a large weight, guiding the model to attend more to those examples. In Proposition 2, We theoretically show that $f_{gnn}$ would converge to learn as well on all discovered topology groups under mild assumptions.

### 3.3 WL-based Auxiliary Task and Final Objective

The proposed TopoImb is model-agnostic and can work with different GNNs. To guide the training of the topology extractor, we further use some auxiliary tasks. For node classification, we assign a pseudo topology label to each node based on its ego-graph (local topology) structures. Concretely, we give each node an initial label based on attributes clustering, and then run WL algorithm on the graph for two rounds to obtain pseudo topology labels as $\mathbb{C}'$. Obtained pseudo labels would carry isomorphism-related features, and this auxiliary task can encourage the extractor to encode topology-discriminative properties. Setting initial labels based on node attributes can enrich the information carried by pseudo topology labels without requiring class labels. For graph classification, as topology is usually distinct across graphs (otherwise two graphs will be the same), we directly use graph class label as the pseudo topology label of each graph, which can guide the topology extractor to capture discriminative topology structures for each class. Concretely, a topology label classifier $g_{aux}$ parameterized by $\rho$ is applied on top of $f_{ext}$, with the training loss:

$$\min_{\phi, \mathbb{T}, \rho} \mathcal{L}_{aux} = \begin{cases} -\sum_{v \in \mathbb{V}} \sum_{c \in \mathbb{C}'} \mathbf{1}(Y_u' == c) \cdot \log(g_{aux}(v)[c]), & \textit{node classification} \\ -\sum_{\mathcal{G} \in \mathbb{G}^L} \sum_{c} (\mathbf{1}(Y_{\mathcal{G}} == c) \cdot \log(g_{aux}(\mathcal{G})[c]), & \textit{graph classification}. \end{cases} \tag{8}$$

**Final Objective Function.** Putting everything together, the overall optimization objective is:

$$\min_{\theta}[(1 - \alpha) \cdot \mathcal{L}_{CE} + \alpha \cdot \max_{\phi, \varphi, \mathbb{T}} \mathcal{L}_{RE}] + \min_{\phi, \mathbb{T}, \rho} \mathcal{L}_{aux} \tag{9}$$

**Training.** An alternative optimization strategy is adopted to solve the objective in Eq. 9. Concretely in each step, we first update parameter $\theta$ of target GNN $f_{gnn}$ with other modules fixed:

$$\min_{\theta} \mathcal{L}_{GNN} = (1 - \alpha) \cdot \mathcal{L}_{CE} + \alpha \cdot \mathcal{L}_{RE}. \tag{10}$$

Then, we fix templates $\mathbb{T}$ and $\theta$ and perform: $\max_{\phi, \varphi} \mathcal{L}_{RE} + \min_{\phi, \rho} \mathcal{L}_{aux}$. Finally, we update templates $\mathbb{T}$ following $\max_{\mathbb{T}} \mathcal{L}_{RE} + \min_{\mathbb{T}} \mathcal{L}_{aux}$. These steps are conducted iteratively until convergence.

### 3.4 Theoretical Analysis

For better understanding of topology imbalance problem, we conduct some analysis on the impact of subclass-level imbalance and the proposed TopoImb. With mild assumptions, a lower bound of performance on a balanced testing set can be derived:

**Table 1:** Imbalanced node classification on three datasets, with the best performance emboldened.

| Method | ImbNode | | | Photo | | DBLP | |
|---|---|---|---|---|---|---|---|
| | MacroF | AUROC | TopoAC | MacroF | AUROC | MacroF | AUROC |
| Vanilla | $78.1_{\pm 0.16}$ | $88.9_{\pm 0.15}$ | $73.4_{\pm 0.13}$ | $52.4_{\pm 0.19}$ | $90.5_{\pm 0.13}$ | $60.9_{\pm 0.18}$ | $86.2_{\pm 0.12}$ |
| OverSample | $78.8_{\pm 0.13}$ | $89.3_{\pm 0.14}$ | $73.7_{\pm 0.15}$ | $56.1_{\pm 0.20}$ | $91.1_{\pm 0.15}$ | $61.5_{\pm 0.17}$ | $84.5_{\pm 0.11}$ |
| ReWeight | $78.9_{\pm 0.16}$ | $89.6_{\pm 0.16}$ | $73.8_{\pm 0.17}$ | $55.8_{\pm 0.20}$ | $88.6_{\pm 0.14}$ | $61.6_{\pm 0.19}$ | $86.6_{\pm 0.12}$ |
| EmSMOTE | $79.6_{\pm 0.13}$ | $90.3_{\pm 0.11}$ | $74.1_{\pm 0.10}$ | $56.5_{\pm 0.17}$ | $92.1_{\pm 0.11}$ | $60.7_{\pm 0.16}$ | $88.6_{\pm 0.10}$ |
| Focal | $77.9_{\pm 0.15}$ | $87.8_{\pm 0.12}$ | $72.5_{\pm 0.12}$ | $53.5_{\pm 0.19}$ | $90.4_{\pm 0.12}$ | $61.4_{\pm 0.18}$ | $83.2_{\pm 0.11}$ |
| GSMOTE | $76.6_{\pm 0.17}$ | $88.2_{\pm 0.16}$ | $73.2_{\pm 0.14}$ | $58.4_{\pm 0.21}$ | $92.3_{\pm 0.14}$ | $63.4_{\pm 0.19}$ | $87.9_{\pm 0.13}$ |
| ReNode | $74.7_{\pm 0.15}$ | $87.9_{\pm 0.13}$ | $72.9_{\pm 0.14}$ | $56.3_{\pm 0.19}$ | $90.7_{\pm 0.13}$ | $61.3_{\pm 0.18}$ | $87.7_{\pm 0.12}$ |
| RECT | $77.2_{\pm 0.14}$ | $89.2_{\pm 0.12}$ | $73.6_{\pm 0.13}$ | $51.2_{\pm 0.18}$ | $91.2_{\pm 0.12}$ | $59.8_{\pm 0.17}$ | $84.1_{\pm 0.10}$ |
| DR-GCN | $78.4_{\pm 0.15}$ | $89.7_{\pm 0.13}$ | $73.7_{\pm 0.15}$ | $57.6_{\pm 0.21}$ | $91.5_{\pm 0.15}$ | $61.9_{\pm 0.16}$ | $87.3_{\pm 0.13}$ |
| TopoImb | $\mathbf{82.1}_{\pm 0.14}$ | $\mathbf{92.3}_{\pm 0.11}$ | $\mathbf{75.2}_{\pm 0.09}$ | $\mathbf{58.9}_{\pm 0.18}$ | $\mathbf{92.7}_{\pm 0.11}$ | $\mathbf{63.9}_{\pm 0.15}$ | $\mathbf{88.7}_{\pm 0.09}$ |

**Proposition 1.** *For the imbalanced training data $\mathcal{D}$ with imbalance ratio $R \triangleq \frac{\tau_{minor}}{\tau_{major}} \geq r$, minimization of empirical loss could result in a hypothesis $\hat{\theta} \in \mathcal{H}$ that is biased towards the majority region $r_{major}$, with its loss bounded as $\epsilon_{test} \leq \epsilon_{train} + 2(\frac{1}{r} - 1) + \lambda^*$ on the ideally balanced test set. $\lambda^*$ is a constant for hypothesis space $\mathcal{H}$.*

It is shown in Proposition 1 that $\epsilon_{test}$ is bounded by imbalance ratio of sub-class regions, corresponds to topology groups in graphs. Next, we further analyze the convergence property of TopoImb:

**Proposition 2.** *If $f_{ext}$ and $f_{gnn}$ has enough capacity, and at each step $f_{gnn}$ is updated w.r.t reweighted loss $\mathcal{L}_{GNN}$, then $f_{gnn}$ converges to have the same error across data regions $r_k$.*

Proposition 2 justifies the effectiveness of TopoImb against topology-level imbalance problem from the theoretical view. Proofs of both two propositions are provided in Appendix B.

## 4 Experiment

We now demonstrate the effectiveness of our proposed TopoImb in handling imbalanced graph learning through experiments on three node classification and three graph classification datasets.

### 4.1 Experimental Settings

**Datasets.** For node classification, we adopt a synthetic dataset, ImbNode, and two real-world datasets: Photo [22] and DBLP [23]. ImbNode is created by attaching two types of motifs, *Houses* and *Cycles*, into a base BA graph [24]. Nodes in the built graph are labeled based on their positions. This design enables explicit control over sub-class imbalance ratios. More details can be found in Appendix A.1. Photo is the Amazon Photo network, with nodes representing goods and edges denoting that two goods are frequently bought together. Labels are set based on the respective product category of each good. DBLP is a citation network with papers as nodes and citations as edges. Nodes are labeled by their research domains. For Photo and DBLP, different topology groups encode different product/paper features, hence they are also selected for the experiments.

For graph classification, we conduct experiments on ImbGraph, and molecular graphs [25] including Mutag and Enzymes. The synthetic dataset ImbGraph is generated as classifying three groups of motifs: *Grid-like structures* (includes Grids and Ladder), *Cycle-like structures* (includes Cycles and Wheels), *Default structures* (includes Trees and Houses). In constructing ImbGraph, we set the size of each motif group as $\{500, 300, 130\}$ respectively, and the ratio of motifs within the same group is $0.2$. More details in creating ImbGraph are provided in Appendix A.2.

Statistics of these datasets can be found in Appendix A.

**Configurations.** In experiments, the Adam optimizer is adopted for all methods, with learning rate initialized to 0.01 and weight decay as 5e-4. All methods except for GSMOTE are trained until convergence with the maximum epoch $1,000$. For GSMOTE, it is additionally pre-trained for $1,000$ epochs on graph auto-encoding task.

**Evaluation Metrics.** Following existing works [10, 11], results are reported in terms of macro F measure (MacroF) and AUROC score [26] as they are robust to class imbalance. MacroF computes

**Table 2:** Performance of imbalanced graph classification task, with the best performance emboldened.

| Method | ImbGraph | | | Mutag | | Enzymes | |
|---|---|---|---|---|---|---|---|
| | MacroF | AUROC | TopoAC | MacroF | AUROC | MacroF | AUROC |
| Vanilla | $68.5_{\pm0.09}$ | $88.7_{\pm0.05}$ | $55.1_{\pm0.07}$ | $54.2_{\pm0.12}$ | $83.3_{\pm0.07}$ | $18.5_{\pm0.10}$ | $53.8_{\pm0.08}$ |
| OverSample | $48.0_{\pm0.08}$ | $90.3_{\pm0.06}$ | $47.9_{\pm0.06}$ | $56.3_{\pm0.10}$ | $71.3_{\pm0.09}$ | $20.3_{\pm0.11}$ | $59.7_{\pm0.09}$ |
| ReWeight | $47.5_{\pm0.08}$ | $89.4_{\pm0.06}$ | $48.2_{\pm0.05}$ | $54.7_{\pm0.11}$ | $81.7_{\pm0.08}$ | $20.0_{\pm0.10}$ | $58.8_{\pm0.09}$ |
| EmSMOTE | $55.9_{\pm0.09}$ | $74.8_{\pm0.07}$ | $48.3_{\pm0.08}$ | $59.7_{\pm0.11}$ | $84.1_{\pm0.10}$ | $12.6_{\pm0.11}$ | $54.7_{\pm0.08}$ |
| Focal | $63.7_{\pm0.06}$ | $88.5_{\pm0.05}$ | $58.7_{\pm0.06}$ | $53.7_{\pm0.10}$ | $80.9_{\pm0.08}$ | $16.7_{\pm0.09}$ | $55.9_{\pm0.07}$ |
| TopoImb | $\mathbf{73.9}_{\pm0.07}$ | $\mathbf{92.2}_{\pm0.04}$ | $\mathbf{61.6}_{\pm0.05}$ | $\mathbf{62.1}_{\pm0.11}$ | $\mathbf{84.7}_{\pm0.08}$ | $\mathbf{20.7}_{\pm0.09}$ | $\mathbf{60.4}_{\pm0.06}$ |

the harmonic mean of class-wise precision and recall, and AUROC trades off the true-positive rate for the false-positive rate. We calculate them separately for each class and report the non-weighted average. For datasets ImbNode and ImbGraph, topology labels are readily available, hence we further report the macro topology-group accuracy (TopoAC) by calculating the mean accuracy of different topology groups, providing a direct evaluation of sub-class level imbalance. Discussion of extra computational cost is provided in Appendix D.

## 4.2 TopoImb for Node Classification

In node classification, ImbNode is constructed to be imbalanced at topology level. For datasets Photo and DBLP, as topology groups are not readily available, we take the step imbalance setting [11]. Concretely, half of the classes are selected as the majority, and the remaining classes are treated as the minority. Sizes of Train/val/test sets are split as $n_{train} : n_{val} : n_{test} = 1 : 3 : 6$. The labeling size of each majority class is set to $\frac{n_{train}}{|\mathbb{C}|}$, and the labeling size for each minority class is set to $R \cdot \frac{n_{train}}{|\mathbb{C}|}$, where $R$ is the preset imbalance ratio. We set $R$ to $0.2$ unless noted otherwise. Further discussion of this split is provided in Appendix A. Two groups of baselines are implemented for comparison: (1) Classical imbalanced learning approaches, including OverSample [7], ReWeight [7], Focal loss [20], and EmSMOTE [27]; (2) Imbalanced node classification strategies, including GraphSMOTE (GSMOTE) [10], ReNode [11], RECT [28] and DR-GCN [29]. GCN [30] is used as the backbone model. Each experiment is randomly conducted for 5 times, and the mean performance is summarized in Table 1.

From Table 1, it can be observed that TopoImb outperforms all baselines with a clear margin on ImbNode and Photo, which proves the effectiveness of the proposed method. ImbNode is imbalanced at the sub-class level by design, and it is shown that most classical imbalanced learning methods, like Focal and Reweight, are ineffective in this setting. Methods for graphs like GSMOTE and ReNode rely on the mechanism of label propagation while neglecting to encode local topology structures, and are also shown to be ineffective. We notice that the improvement in DBLP is smaller compared to the other datasets. The reason could be that topology structures are less-discriminative towards node labels in citation networks, rendering topology-group-wise modulation less helpful.

## 4.3 TopoImb for Graph Classification

In graph classification, we again take the step imbalance setting and select half classes as the minority while others as the majority on dataset Mutag and Enzymes, with imbalance ratio $R$ being set to $0.1$. Dataset ImbGraph is constructed to be imbalanced. As graphs can be taken as i.i.d instances in this task, we implement a set of popular imbalanced learning methods: Over-sampling, Re-weight, Embed-SMOTE, and Focal loss. $5\%$ of graphs are used in training, $30\%$ in validation, and $60\%$ in testing. Each experiment is randomly conducted for 5 times with performance reported in Table 2.

As shown in Table 2, TopoImb achieves the best performance across all three datasets, which further validates its effectiveness and generalizability. Topology groups are more difficult to capture in graph-level tasks due to graph isomorphism, but TopoImb is able to discover similar graph groups and modulates training of target GNNs.

## 4.4 Can TopoImb Effectively Modulate the Training Process?

In order to successfully guide the learning process, TopoImb should assign a larger weight to minority or more difficult topology groups, and its behaviors should evolve along with the update of the target

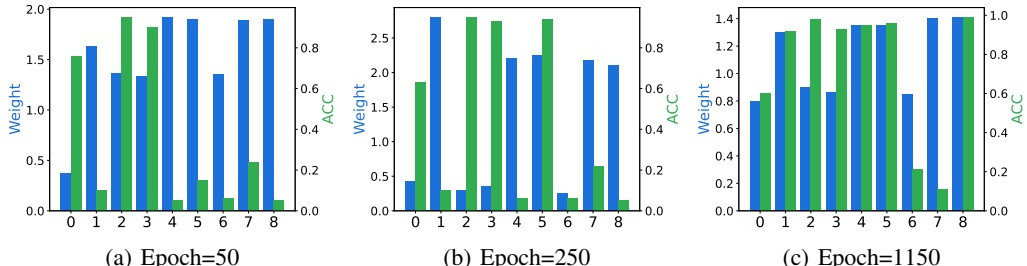

(a) Epoch=50       (b) Epoch=250       (c) Epoch=1150

**Figure 3:** Distribution of assigned weights and mean accuracy of each topology group on ImbNode, at different learning stages.

**Table 3:** Test on other GNN architectures: GraphSage and GIN.

| Method | ImbNode | | | Mutag | | ImbGraph | | |
|---|---|---|---|---|---|---|---|---|
| | MacroF | AUROC | TopoAC | MacroF | AUROC | MacroF | AUROC | TopoAC |
| GraphSage | 59.7 | 82.4 | 57.9 | 51.4 | 81.3 | 67.3 | 88.5 | 57.1 |
| +TopoImb | 63.5 | 86.2 | 59.8 | 55.6 | 82.7 | 72.1 | 91.6 | 59.8 |
| GIN | 65.8 | 87.8 | 60.1 | 68.3 | 91.1 | 97.4 | 99.2 | 97.3 |
| +TopoImb | 74.2 | 92.6 | 64.6 | 79.7 | 91.4 | 99.2 | 99.6 | 99.8 |

GNN model. To evaluate it, we examine the distribution of predicted weights **w** at different training stages in this section. Experiments are conducted on ImbNode which contains explicit topology labels, and other settings are the same as Sec. 4.2. We train a TopoImb model and visualize the mean weights assigned to topology groups as well as their mean accuracy at $\{50, 250, 1150\}$ epochs along the learning trajectory. Results are presented in Fig. 3, and group $\{5, 6, 7, 8\}$ are minority groups. It can be observed that: (1) Minority groups or groups with a low accuracy tend to receive a large weight; (2) In the early stage, the gap of assigned weight scales across topology groups is large; (3) In the late stage, both assigned weights and group-wise performances show a smaller gap. These results further validate the effectiveness of TopoImb.

### 4.5 Do Different Topology Groups Select Different Templates?

TopoImb will automatically learn a template set **T** to encode rich topology information. To analyze its effectiveness, we further check the activation distribution of template selection in this part.

If successfully learned, nodes with different ego-graph structures would attend to different templates, while nodes of the same group would exhibit similar template selection behaviors. Again, we experiment on ImbNode as it has available topology labels, and visualize the distribution of template selections for each topology group in Fig. 4. In Fig. 4, the Y axis is the topology group and the X axis is the template set. It can be observed that generally, different topology groups tend to select different templates.

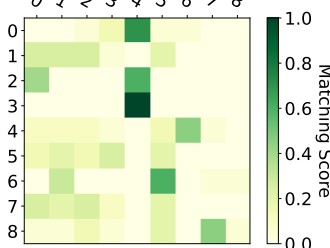

**Figure 4:** Distribution of template selection behaviors.

### 4.6 Is TopoImb Generalizable?

Our proposed TopoImb should be model-agnostic and effective across different architectures of target GNNs. To evaluate its generalizability, we further test it on two other popular GNN models: GraphSage [31] and GIN [15]. We conduct experiments on ImbNode, Mutag, and ImbGraph, with results summarized in Table 3. It can be observed that TopoImb remains effective for both models across these datasets, which verifies the generalizability of TopoImb.

### 4.7 Ablation Study of Reweighter

In this section, we conduct ablation study to evaluate importance of designed topology extractor. Concretely, we compare it with two baselines: (1) Class-wise reweighter, (2) GCN-based reweighter. The first baseline assigns a uniform weight to all examples inside the same class, while the second

one directly use a GCN model as the weight assigner, assigning example weight at instance-level. Both baselines are updated following Eq. 7, by maximizing training loss in the adversarial manner. Experiments are conducted on ImbNode, Photo and DBLP, and keep the same configuration as Sec. 4.2. Results reported in Table. 4.

**Table 4:** Performances of different re-weighter structures.

| Method | ImbNode | | | Photo | | DBLP | |
|---|---|---|---|---|---|---|---|
| | MacroF | AUROC | TopoAC | MacroF | AUROC | MacroF | AUROC |
| Class-wise | 78.1 | 90.4 | 72.4 | 54.2 | 90.5 | 61.5 | 85.4 |
| GCN-based | 79.6 | 92.2 | 73.3 | 57.4 | 91.8 | 61.8 | 85.4 |
| Topo-based | **82.1** | **92.3** | **74.9** | **58.9** | **92.7** | **63.9** | **88.7** |

Results from comparing to class-wise reweighter can show the improvements in considering sub-class-level imbalance. As shown in the table, both GCN-based reweighter and our proposed Topo-based reweighter outperform this baseline consistently. On the other hand, comparison with GCN-based reweighter validates the benefits of explicitly considering latent topology groups during training modulation. As shown in Table. 4, the proposed Topo-based reweighter outperforms GCN-based reweighter with a clear margin, across all three datasets. With the proposed topology extraction module, TopoImb latently regularizes importance assignment at topology group level and can adapt to the evolvement of target GNN with high data-efficiency. While GCN-based reweighter works at instance-level, which could be difficult to give informative weights due to complexity of inputs and unstable prediction of target GNN model for each instance.

### 4.8 Hyperparameter Sensitivity Analysis

We also conduct a series of sensitivity analyses on the number of memory cells and hyper-parameter $\alpha$. Due to the space limit, we put it in Appendix C.

## 5 Related Work

**Graph Neural Networks.** In recent years, various graph neural network models have been proposed for learning on relational data structures, including methods based on convolutional neural networks [30, 32, 33], recurrent neural networks [34, 35], and transformers [36, 37]. Despite their differences, most GNN models fit within the message passing framework [16]. In this framework, node representations are iteratively updated with a differentiable aggregation function that considers features of their neighboring nodes. For instance, GCN [30] adopts the fixed weights for neighboring nodes in its message-passing operation. GAT [19] further introduces the self-attention mechanism to learn different attention scores of neighborhoods. Ref [38–41] propose to augment GNNs with explicit prototypical representations which can increase the data efficiency and model the hierarchical motif distribution. Ref [42–44] propose to disentangle the given graph to uncover latent groups of nodes or edges. More variants of GNN architectures can be found in a recent survey [45]. GNN models have achieved remarkable success in a wide range of graph mining tasks [15, 31, 43, 46, 47]. And recently, works have also been conducted over its trustworthiness [48–51] and explainability [52, 53]. However, despite the popularity, most existing GNNs are built under balanced data-splitting settings. The data imbalance problem appears frequently in real-world applications and could heavily impair the performance of GNN models [10, 54].

**Imbalanced Learning.** Previous efforts to handle the data imbalance problem can be mainly categorized into two groups: data re-sampling [7, 14, 55, 56] and cost-sensitive learning [8, 9, 20, 57–59]. A comprehensive literature survey can be found in [60, 61]. Re-sampling methods adopt either random under-sampling or oversampling techniques to obtain a balanced distribution. The vanilla over-sampling method simply duplicates underrepresented samples. However, this method may cause the overfitting problem due to repeating training on duplicated samples without extra variances. To alleviate this problem, SMOTE [55] generates new training samples by interpolating neighboring minority instances. Various extensions are then proposed with more sophisticated interpolation

processes [10, 56, 62, 63]. Instead of manipulating the input data, cost-sensitive learning methods operate at the algorithmic level by imposing varying error penalties for different samples. A manner to design the weighting function is to assign larger weights to samples with larger losses, including boosting-based algorithm [64, 65], hard example mining [66], and focal loss [20]. Considering that the prior knowledge may be unavailable in real problems, Meta-Weight-Net [67] parameterizes the weighting function with an MLP (multilayer perceptron) network to adaptively learn a weighting function from data.

More recently, some efforts have been made to improve imbalanced node classification [10, 11, 68–71] and graph classification [72, 73]. For instance, GraphSMOTE extends SMOTE to deal with graph data [10]. Mixup is introduced to improve imbalanced node classification in [68, 74]. DPGNN proposes a class prototype-driven training loss to maintain the balance of different classes [71]. To alleviate the overfitting and underfitting problems caused by lacking sufficient prior knowledge, GNN-CL proposes a curriculum learning framework with an oversampling strategy based on smoothness and homophily [75]. Ref [11, 76] propose to consider a special type of node imbalance in terms of their positions in the graph. However, the fine-grained sub-class data imbalance problem inherent in real-life graphs has been less explored in the literature. Instead, our method emphasizes underrepresented sub-class groups and makes the first attempt to tackle the imbalanced graph learning problem in a more general setting.

## 6    Conclusion

In this work, we consider a critical challenge: imbalance may exist in sub-class topology groups instead of pure class-level. Most existing methods rely on knowledge of imbalance ratios or require explicit class splits, hence are ineffective for this problem. A novel framework TopoImb is proposed, which can automatically discover under-represented groups and modulate the training process accordingly. Theoretical analysis is provided on its effectiveness, and experimental results on node classification and graph classification tasks further validate its advantages.

As a future direction, more effective modulation mechanisms can be explored. Currently, we limit it to assigning different weights towards training instances to keep simplicity. In the future, novel modulation techniques that can augment the dataset, provide extra knowledge, or directly manipulate target models can be designed, which may further boost the performance.

## 7    Acknowledgement

This material is based upon work supported by, or in part by, the National Science Foundation under grants number IIS-1707548 and IIS-1909702, the Army Research Office under grant number W911NF21-1-0198, and Department of Homeland Security CINA under grant number E205949D. The findings and conclusions in this paper do not necessarily reflect the view of the funding agency.

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

# A  Datasets Description

In this work, we test TopoImb on six datasets, three for node classification and three for graph classification. Statistics of these datasets are summarized in Table 5. Next, we will go into details about the generation of two synthetic datasets, ImbNode and ImbGraph.

**Table 5:** Statistics of graph datasets used in this work.

| Name | #Graphs | #Nodes | #Edges | #Features | #Classes |
|---|---|---|---|---|---|
| ImbNode | 1 | $1,592$ | $6,192$ | 10 | 4 |
| Photo | 1 | 7,650 | 238,162 | 745 | 8 |
| DBLP | 1 | 17,716 | 105,734 | 1,639 | 4 |
| ImbGraph | 2997 | $\sim 27.6$ | $\sim 117.8$ | 10 | 3 |
| Mutag | 188 | $\sim 17.9$ | $\sim 39.6$ | 7 | 2 |
| Enzymes | 600 | $\sim 32.6$ | $\sim 124.3$ | 3 | 6 |

## A.1  ImbNode

ImbNode is a dataset for node classification with intrinsic sub-class-level imbalance, in which topology information of each node is explicitly provided. In constructing ImbNode, we use two groups of motifs, *Houses* and *Cycles*, and a base BA graph. Each node is labeled based on their positions, and a topology label is also provided to differentiate each topology group. Nodes in the BA graph is labeled as $0$ for both class and topology groups. Labels of nodes in motifs are set as shown in Fig. 5. Each motif has $5$ nodes, and nodes with the same topology label have the same 2-hop ego graph structure. Concretely, we built $166$ Houses and $33$ Cycles, and randomly attach them to the base BA graph of $597$ nodes.

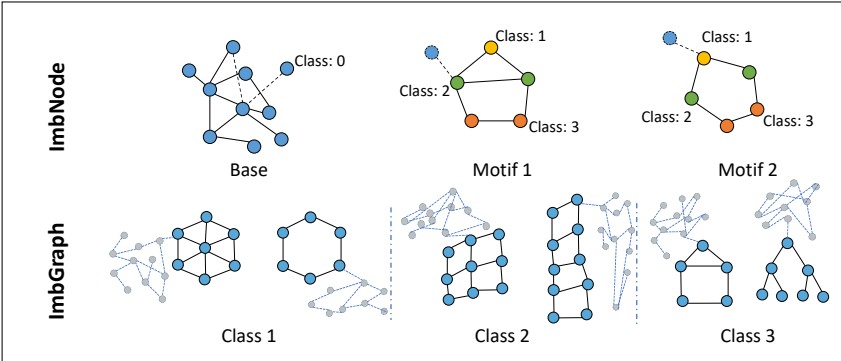

**Figure 5:** Examples of synthetic datasets, ImbNode and ImbGraph. In ImbNode, color denotes the class label. With WL-algorithm, nodes in motifs can be clustered into $8$ topology groups. In ImbGraph, we show one representative example for each topology group inside three classes. Nodes in doted line are the random BA graph.

## A.2  ImbGraph

ImbGraph is created for graph classification, with each class containing several distinct topological structures. Concretely, we have three classes: *Grid-like motifs* (includes Grids and Ladder), *Cycle-like motifs* (includes Cycles and Wheels), and *Default motifs* (includes Trees and Houses). For each class, examples are created by attaching the corresponding motif to a random BA graph. The size of each class is set to $\{500, 300, 130\}$ respectively, and topology groups of each class is also imbalanced with imbalance ratio $0.2$. Examples are provided in Fig. 5. Graphs have topology labels readily available, enabling us to directly evaluate model performance w.r.t sub-class-level imbalance.

## A.3  Real-world Node Classification Datasets

For real-world node classification datasets, Photo and DBLP, we split the training data by varying ratio of each class as in Sec. 4.2. Note that this data split does not directly manipulate the data distribution in the topology level. The reason of adopting this setting is two-folded: (1) topology labels of nodes are unavailable in these real-world graphs, making it difficult to directly create topology imbalance

scenarios. (2) Although not explicitly split to be topology imbalanced, we expect that such imbalance property exists in obtained dataset, and use experiments to show that our TopoImb does perform better in this configuration. This proxy data-split strategy is also adopted by [1], which works on a special type of imbalance by considering the position of nodes in the graph.

To further validate our data split and justify the existence of topology imbalance, we conduct an analysis by clustering the embeddings obtained with our topology extractor (which has guaranteed topology expressiveness as WL-test). Concretely, we first obtain the embeddings of all nodes in dataset Photo which encode their topology information. Then, for each class we cluster them with MeanShift to get sub-class groups. Finally, we calculate the probability of training examples from current class to fall into these groups . This analysis verify the existence of sub-class topology imbalance in our scenario. Taking the first three classes of dataset Photo as an example, we can obtain 6 groups for each of them with MeanShift and the distribution of these classes w.r.t those topology groups are $[0.54, 0.14, 0.11, 0.09, 0.08, 0.04]$, $[0.63, 0.19, 0.08, 0.04, 0.03, 0.03]$ and $[0.55, 0.17, 0.09, 0.09, 0.05, 0.05]$ respectively. It is shown that sub-class level imbalance exists in these groups.

# B    Theoretical Analysis

In this section, we provide some analysis of our proposed TopoImb in learning from sub-class-level imbalanced graphs. Following the discussion in 2.1, we conduct the analysis on a simplified setting, in which the distribution of training data $\mathbb{D}$ can be split into $K$ regions: $p_{\mathbb{D}}(\boldsymbol{x}) = \sum_{k=1}^{K} p_{r_k}(\boldsymbol{x}) \cdot \tau(r_k)$. $\boldsymbol{x}$ represents the training example, each region $r_k$ is a topology group, and $\tau(r_k) \geq 0$ denotes its distribution density with $\sum_{k=1}^{K} \tau(r_k) = 1$. As $\mathbb{D}$ is imbalanced, variance of $\tau(r_k)$ is large, e.g., $\tau(r_{major}) \gg \tau(r_{minor})$. Without loss of generality, $\boldsymbol{x}$ can also represent data embeddings, increasing generalizability of our analysis.

## B.1    Proof of Proposition 1

*Proof.* First, the expected empirical loss can be calculated as:

$$\epsilon_{train} = \sum_{k=1}^{K} \tau(r_k) \cdot \mathbb{E}_{\boldsymbol{x} \sim p_{r_k}} \sum_{c} \mathbf{1}(Y_{\boldsymbol{x}} == c) \cdot \log(P_{\boldsymbol{x}}[c]) \tag{11}$$

For the ideally balanced case, we have $\tau(r_k)^* = \frac{1}{K}$ and:

$$\epsilon_{test} = \sum_{k=1}^{K} \tau(r_k)^* \cdot \mathbb{E}_{\boldsymbol{x} \sim p_{r_k}} \sum_{c} \mathbf{1}(Y_{\boldsymbol{x}} == c) \cdot \log(P_{\boldsymbol{x}}[c]) \tag{12}$$

Note that $\tau(r_k)$ usually deviates significantly from $\tau(r_k)^*$ due to the existence of graph imbalance. Clearly, comparing Eq. 11 and Eq. 12, empirical error would emphasize regions with a high distribution density, and neglect those whose $\tau$ is small, resulting in hypothesis $\hat{\theta}$ biased towards majority regions.

With the given data distribution $p_{\mathbb{D}}(\boldsymbol{x}) = \sum_{k=1}^{K} p_{r_k}(\boldsymbol{x}) \cdot \tau(r_k)$ and ideally balanced data distribution $p_{\mathbb{D}^*}(\boldsymbol{x}) = \sum_{k=1}^{K} p_{r_k}(\boldsymbol{x}) \cdot \frac{1}{K}$, we can further derive the testing error bound. From [77], we have:

$$\epsilon_{test} \leq \epsilon_{train} + d_{\mathcal{H}}(p_{\mathbb{D}}, p_{\mathbb{D}^*}) + \lambda^*, \tag{13}$$

where $\lambda^*$ is the optimal joint error on both distributions, and is a constant. $d_{\mathcal{H}}(p_{\mathbb{D}}, p_{\mathbb{D}^*})$ measures $\mathcal{H}$-divergence across two distributions. Assuming data distribution at each region $r_k$ are i.i.d, we can

get:

$$
\begin{aligned}
d_{\mathcal{H}}(p_{\mathbb{D}}, p_{\mathbb{D}^*}) &\triangleq 2 \sup_{h \in \mathcal{H}} |Pr_{\boldsymbol{x} \sim \mathbb{D}}[h(\boldsymbol{x}) = 1] - Pr_{\boldsymbol{x} \sim \mathbb{D}^*}[h(\boldsymbol{x}) = 1]| \\
&= 2 \sup_{h \in \mathcal{H}} |\sum_{k=1}^{K} Pr_{\boldsymbol{x} \sim r_k}[h(\boldsymbol{x}) = 1] \cdot \tau(r_k) - \sum_{k=1}^{K} Pr_{\boldsymbol{x} \sim r_k}[h(\boldsymbol{x}) = 1] \cdot \tau(r_k)^*| \\
&= 2 \sup_{h \in \mathcal{H}} |\sum_{k=1}^{K} Pr_{\boldsymbol{x} \sim r_k}[h(\boldsymbol{x}) = 1] \cdot (\tau(r_k) - \tau(r_k)^*)| \\
&\leq 2 \sum_{k=1}^{K} |\tau(r_k) - \frac{1}{K}| \\
&\leq 2 \sum_{k=1}^{K} (\frac{1}{K \cdot r} - \frac{1}{K}) \\
&= 2(\frac{1}{r} - 1),
\end{aligned}
\tag{14}
$$

which concludes the proof. $\qquad\square$

From this result, we can observe that with increased imbalance (lower $r$), range of $\epsilon_{test}$ will also increase, and performance of trained $f_{gnn}$ will become less-guaranteed on the balanced test set.

Then, we can construct the connection between our proposed TopoImb and achieving a lower testing loss in Eq. 12. Intuitively, TopoImb is able to re-weight instances from each region, change Eq. 11 by assigning larger importance to under-learned groups. Typically, minority regions would be up-weighted and majority regions would be down-weighted, which would decrease the imbalance in effect, and provide a better guarantee on testing performance, as shown in Proposition 1.

### B.2 Proof of Proposition 2

*Proof.* Let $\mathcal{L}_{r_k} = \mathbb{E}_{\boldsymbol{x} \sim p_{r_k}} \sum_c (\mathbf{1}(Y_{\boldsymbol{x}} == c) \cdot \log(P_{\boldsymbol{x}}[c])$, which represents error of $f_{gnn}$ in region $r_k$. As shown in Eq. 9, the learning function of TopoImb can be summarized as:

$$
\min_{P_{\boldsymbol{x}}} \max_{\boldsymbol{w}} \sum_{k=1}^{K} (1 + \boldsymbol{w}[k]) \cdot \tau(r_k) \cdot \mathcal{L}_{r_k}
\tag{15}
$$

where $P_{\boldsymbol{x}} \sim f_{gnn}(\boldsymbol{x})$, $\boldsymbol{w} \in \mathbb{R}^K$ with each element in $[0, 1]$. At each step with $P_{\boldsymbol{x}}$ fixed, $\boldsymbol{w}$ is updated to maximize $\sum_{k=1}^{K} (1 + \boldsymbol{w}[k]) \cdot \tau(r_k) \mathcal{L}_{r_k}$. It is a convex linear combination problem, and a larger weight would be given to regions with higher errors after update.

Note that for any given $\boldsymbol{w}$, $\min_{P_{\boldsymbol{x}}} \sum_{k=1}^{K} (1 + \boldsymbol{w}[k]) \cdot \tau(r_k) \cdot \mathcal{L}_{r_k}$ is convex in $P_{\boldsymbol{x}}$. It is known that subderivatives of a supremum of convex functions include the derivative of the function at the point where the maximum is attained [21]. That is, given $\boldsymbol{w}' = \arg\max_{\boldsymbol{w}} f_{obj}$, $\partial f_{obj, \boldsymbol{w}=\boldsymbol{w}'} \in \partial f_{obj}$, $P_{\boldsymbol{x}}$ would converge to its optima. Furthermore, $\boldsymbol{w}$ would not converge until each region has the same error. Therefore, with sufficiently small updates of $P_{\boldsymbol{x}}$, $P_{\boldsymbol{x}}$ would converge, at each point $\mathcal{L}_{r_i} = \mathcal{L}_{r_j}, \forall i, j \in [0, \dots, K]$. $\qquad\square$

This result shows the convergence property of TopoImb, and verifies that it will guide the learning of $f_{gnn}$, preventing the problem of under-representation of minority regions.

## C  Sensitivity Analysis

We also conduct a series of sensitivity analyses on the number of memory cells and hyper-parameter $\alpha$, which controls the weight of $\mathcal{L}_{RE}$. Experiments are conducted on a node classification dataset, Photo, and a graph classification, Enzymes. For the analysis on memory cells, we vary its size in $[2, 4, 6, 8, 10, 12, 14]$, and all other settings remain unchanged. We run each experiment randomly for three times and report the average results in terms of AUROC score in Fig. 6[a-b]. It can be

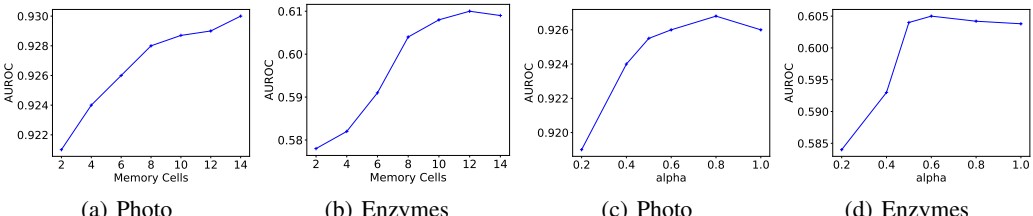

|                |                |                |                |
| :------------: | :------------: | :------------: | :------------: |
| (a) Photo      | (b) Enzymes    | (c) Photo      | (d) Enzymes    |

**Figure 6:** Sensitivity analysis. [a-b] show the influence of memory cells number, and [c-d] show the influence of hyper-parameter $\alpha$. Results are reported in terms of AUROC score.

observed that (1) the performance would increase with the number of memory cells generally, (2) when the number is larger than $8$, further increasing it may only slightly improve the performance.

Similarly, for the analysis on $\alpha$, we vary it within $[0.2, 0.4, 0.5, 0.6, 0.8, 1]$ without changing other settings. Average results of three random running are reported in Fig. 6[c-d]. It can be observed that both dataset would benefit from a large $\alpha$ within the scale $[0, 0.5]$. When $\alpha$ is larger than $0.8$, increasing it further may result in a performance drop. As a large $\alpha$ would under-weight the original classification loss in Eq. 10, it may result in more noises in the training process, which could be the possible reason behind this phenomenon.

## D Computational Cost

In this part, we discuss the extra computational cost introduced by TopoImb.

- First, we compare the inference time between our model and the vanilla model which has the topology extractor removed. To reduce the variance, we randomly run for $100$ times and report the total time cost. Comparisons between the vanilla model and our TopoImb are as follows: ImbNode ($1.38s$ vs $1.43s$); Photo ($1.36s$ vs. $1.45s$); DBLP ($1.50s$ vs $1.57s$); ImbGraph ($0.29s$ vs $0.53s$); and Enzymes ($0.54s$ vs $1.15s$). It can be seen that the proposed TopoImb would not introduce a significant increase in term of computation time. Besides, the extra computation time is a little higher on the graph classification task. This phenomenon could arise due to the difference in graph sizes and the attentive pooling layer for graph representations.

- Second, we analyze the model sizes by comparing parameters of different networks. Concretely, on dataset Photo, we calculate the parameter sizes of different backbone GNNs and our designed topology extractor, which are $15,668$ (2-layer GCN), $16,508$ (2-layer GIN), $30,968$ (2-layer Sage) and $18,289$ (topology extractor) respectively. Note that when a more complex GNN is adopted, the influence of the additional reweighter module would become smaller.

- Last, for the optimization time, due to the alternative optimization algorithm used in TopoImb, it is slower than adopting the backbone network alone. For example on node classification dataset DBLP, the training time (of $1,000$ epochs) increases from 16.7 seconds to 44.7 seconds when a 2-layer GCN is used as the backbone. Despite the increase, it can be seen that optimization time would still not be a major problem for our proposed framework. Furthermore, note that there is no extra computation cost in testing after the model is trained.

