# OpenReview forum: "TopoImb: Toward Topology-level Imbalance in Learning from Graphs"
_logconference.io/LOG/2022/Conference — LoG 2022 Poster_

### Official Review · Reviewer_NA3x · 2022-09-25

**Overall Score:** 6
**Confidence:** 5

**Review:**

Strength:
1. This paper focuses on a new problem, the sub-class topology level imbalance problem, which is seldom noticed by the community and needs more attention.
2. This paper proposed a model-agnostic method and the proposed method seems to be effective on the evaluation datasets.

Weakness:
1. My major concern is that what’s the sub-class topology level imbalance for node classification. According to the dataset separation of Photo and DBLP in section 4.2, I think it’s more like a quantity imbalance rather than a topology imbalance.
2. Since it’s the first work that analyzes the sub-class level imbalance problem in graph learning, there should be a formal problem definition. I think the definition may be different for the graph classification and node classification tasks.
3. The proposed template is very similar to the graph prototype[1][2] and the graph disentanglement[3]. These works are missing in the references and in the baselines.
4. The representation of this paper is not very well. I can't clearly see the motivation and the rationale of the proposed topology extractor.
5. More detailed experimental settings of the proposed method and other baselines are needed.

Other comments and questions:
1. What are the limitations of the proposed method? Since section 2.1 assume that a class contains multiple prototypes, are there some assumptions on the datasets?
2. I found that both the ReNode and PASTEL[4] define the topology-imbalance problem for graph semi-supervised classification which is different from this paper. The key difference should be analyzed.
3. In Fig.4, the templet weights of class 8 are nearly 0. I think it needs some explanation.
4. I think it’s better to verify the generalization of the proposed method on different types of datasets in section 4.6 except for the GNN encoder.
5. The number of template K seems to be one of the most important hyper-parameters. The sensitivity of K is needed.
6. The paper needs further proofreading. For example, “macroF” in Table 2 and “MacroF” in Table 3.

[1]Lin S, Liu C, Zhou P, et al. Prototypical graph contrastive learning[J]. IEEE Transactions on Neural Networks and Learning Systems, 2022.
[2]Wang Z, Wang J, Guo Y, et al. Zero-shot node classification with decomposed graph prototype network[C]//Proceedings of the 27th ACM SIGKDD Conference on Knowledge Discovery & Data Mining. 2021: 1769-1779.
[3]Yang Y, Feng Z, Song M, et al. Factorizable graph convolutional networks[J]. Advances in Neural Information Processing Systems, 2020, 33: 20286-20296.
[4]Sun Q, Li J, Yuan H, et al. Position-aware Structure Learning for Graph Topology-imbalance by Relieving Under-reaching and Over-squashing[J]. arXiv preprint arXiv:2208.08302, 2022.

---

### Official Review · Reviewer_Egzp · 2022-10-23

**Overall Score:** 8
**Confidence:** 4

**Review:**

This paper studies the problem of imbalanced topological structures in graph and node classification tasks. The idea is that the distribution of network motifs is rarely uniform and this often leads GNNs to ignore discriminative but infrequent structures. To remedy this problem, this work presents a method to upweigh uncommon topological motifs. The approach (TopoImb) introduces a GNN enhanced with external memory that groups similar topologies and assigns higher weights to minority motif instances.

Strengths:
* This work poses that uncommon yet discriminative structures tend to be ignored by GNNs, providing a unique and under-explored view on the problem of topological motif imbalances.
* Despite the challenging task of identifying topological structures, the proposed approach is fairly simple and rooted in classical algorithms (e.g. AdaBoost) that put more weight on hard-to-classify samples. The adversarial learning idea is conceptually appealing.
* The experiments are fairly comprehensive (8 baselines, 6 datasets, 2 tasks, 3 metrics with error bars). The proposed approach convincingly outperforms the baselines.

Weaknesses and questions:
* What is the rationale behind the auxiliary loss presented in Section 3.3? Why is this loss term necessary and how does TopoImp perform without this term?
* How did the authors optimise the baseline methods? For the results reported in Table 1, did they use the same backbone GNN for all baselines? What does the vanilla method represent? Does it correspond to the TopoImb approach without the topology extractor component? The code to reproduce the experiments is not available. The employed hyperparameters are not reported.
* While the motivation for using the topology extractor to learn and match different templates make sense conceptually, it is unclear whether the method behaves as expected. What kind of templates is the feature extractor learning? In Figure 4, it seems that the 4th template matches 3 different topology groups, while none of the group matches the 8th template. What could be the reason for this?

Minor comments:
* In the abstract, the sentence “the imbalance is likely to exist at the sub-class topology group level” is slightly unclear. The precise meaning of “sub-class imbalances” only becomes apparent in the introduction when these imbalances are presented as infrequent topological motifs.
* Figure 3 is compelling. However, for some reason the weight for the 6th topology group is quite small in all three figures. Why is that the case, considering that the model tends to mis-classify samples with that topology group?
* In Section 4.3, why are only 5% of graphs used in training? It is stated that “5% of graphs are used in training, 30% in validation, and 60% in testing“. Why did the authors discard 5% of the graphs?
* Equations 4, 6, 7, 11, 12: Unclosed parenthesis “(“
* Table 3: “GraphSasge”
* Section 4.6: Active/passive voice consistency: “An ablation study is conducted …” vs “we experiment on …”
* Line 342: effectiveness=

In the light of the unique view on topological motif imbalances and the conceptually appealing approach, I recommend a weak accept for this paper. I will happily raise my score provided that my concerns are successfully addressed.

UPDATE: All my concerns have been addressed. I've raised my score and I now recommend the acceptance of this paper.

---

### Official Review · Reviewer_EPcV · 2022-10-24

**Overall Score:** 6
**Confidence:** 4

**Review:**

### Paper Summary
This paper proposes TopoImb, a novel architecture that seeks to attend to and modulate topology-level imbalance in graph representation learning. The proposed method is modular and effective in node-level and graph-level classification by improving performance in addressing data imbalance.

### Strengths
* topology-imbalance in graph learning is an under-explored problem and opens up a rich space of architectural design. Real-world graphs are often imbalance in terms of topology structures
* the paper is well illustrated, well motivated and well written.
* extensive experiments and decent results.

### Weaknesses
* lack of experiments and analysis with respect to topology-level imbalance: e.g. it is unclear to me how the better ability to address topology-level imbalance is reflected in ~4% increase in AUROC in Table 1.
* ImbNode and ImbGraph are not available even though they are used for evaluation.

### Recommendation
Based on the novelty of the proposed method, I recommend weak acceptance. My main concerns are around the lack of analysis on the computational efficiency and qualitative aspects of topology groups.

### Questions
* what are, qualitatively, the topological groups 0-8 in figure 3? It is hard to reason about this behavior when there’s no analysis on the structure or each and why (if) they are hard to learn / imbalanced in ImbNode.
* the topology extractor does not seem to come for free. What is the computational cost of running this on top of existing GNN in terms of additional run-time/parameters?

### Additional Feedbacks
* there should be more space between the caption and the graphics of figure 1.

### Paper Type
The paper type is correct.

---

### Official Review · Reviewer_n641 · 2022-10-26

**Overall Score:** 8
**Confidence:** 4

**Review:**

**General**

In this paper, the authors address the crucial problem of learning from an imbalanced graph dataset. While most existing methods try to solve this problem by considering the class-level imbalance, this work provides a solution that considers a sub-class topology group imbalance. The solution (TopoImb) is a framework consisting of a topology extractor and training modulator. It can be used to train any GNN. The results on both artificial and real-world datasets in terms of node and graph classification tasks show that the proposed framework is able to boost the overall performance of the trained GNN.


**Strengths**

S1) The problem setting (learning from imbalanced graph datasets) is crucial in real-world applications. The paper is generally well-written (I suggest another proof-read round to eliminate typos and grammar issues) and easy to follow.

S2) The choice of all components of the proposed TopoImb framework is well-motivated, and the description is detailed enough. The authors also provide a theoretical analysis of the topology imbalance problem.

S3) The experimental scenario is extensive, covering both node and graph classification tasks and a broad ablation study.


**Weaknesses**

W1) [*Performance gain vs training time*]

For the node classification task, the gain on real-world datasets is much smaller than on the artificial one; however, the authors state "that TopoImb outperforms all baselines with a clear margin on ImbNode and Photo, (...)". In particular:

MacroF: ImbNode: 82.1% vs 79.6% (2.5pp gain; EmSMOTE), Photo: 58.9% vs 58.4% (0.5pp gain; GraphSMOTE), DBLP: 63.9% vs 63.4% (0.5pp gain; GraphSMOTE)

AUCROC: ImbNode: 92.3% vs 90.3% (2pp gain; EmSMOTE),   Photo: 58.9% vs 58.4% (0.4pp gain; GraphSMOTE), DBLP: 88.7% vs 88.6% (0.1pp gain; EmSMOTE)

- Can this be related to the data preprocessing?
- Although TopoImb receives the overall best metric values, I am wondering how these performance gains relate to the training time/complexity (e.g., comparing the actual training time and the number of parameters).
- It would be great if the authors could provide more details on the training setups for the competitive methods (baselines; including the number of epochs / estimated training time). In fact, the training budget should be similar to provide a fair comparison.

W2) [*Relation of learned templates to existing topological features*]

In Section 4.5, the authors provide an experiment that shows which templates are selected by nodes from different topology groups (in the artificial dataset).

- It would be nice to visualize somehow what the learned templates look like / correspond to.
- However, what's more important here: do these templates resemble well-known topological graph features? (E.g., Betti numbers, cycles, triangles, motifs).
- How can we ensure that the learned templates correspond to topological features, not just "random" vectors?
- A possible extension to the ablation study is to replace the learnable template extractor with a set of predefined topological graph features.

W3) [*No code available*]

Reproducibility is crucial in modern machine learning research. I did not find any code linked in the paper or attached on OpenReview. I encourage the authors to provide their implementation of the presented experimental evaluation, both in terms of code for the proposed TopoImb framework, as well as all (major) experimental tasks (node and graph classification), including baselines.


**Other remarks**

R1) Section 2.1. Do we need a supervised setting? Can't we pretrain the GNN in an unsupervised manner, while preserving the topological balance? Could the topological extractor be pretrained?

R2) Section 3.1. Eq. (3) what operation is denoted here by `\cdot`? Matrix multiplication? If I understand correctly the new node embedding is a weighted summation of the topological templates, right? Is this step always performed in your framework? The sentence suggests that it is an optional step "would be updated" - please consider rephrasing it.

R3) Section 3.3. How to choose the number of clusters for the initial clustering before WL?

R4) Figure 1 - What should the ideal decision boundary look like? In an unbalanced dataset, the majority class will have the largest impact of the classifier's decision boundary, and we should pay more attention to less frequent instances. This claim is not reflected in the figure - we just notice that there are minority example within a class, but the decision boundary is still fine.


**Typos/grammar**
Below I listed some typos/grammar issues that caught my attention, but I would suggest the authors should proof read the paper again to eliminate other uncaught typos.

- L12: missing closing parenthesis - `design (1 a topology` -> `design: (1) a topology`
- L13: missing closing parenthesis - `cells, (2 a training` -> `cells, (2) a training`
- L36: shouldn't it be `inputs`?
- L43: `explicitly`
- L64: `provide a theoretical analysis` (missing `a`)
- L125: `consisting of`
- L144: `the training of a GNN` or `the training of GNNs` (depending on the intent)
- L153: `is the embedding of node (...) and the embedding of graph`
- L267: `TopoImb would automatically learn (...)` - didn't you mean: `will automatically`?
- L285: `Strucutre` -> `Structure`
- L342: remove `=`
- L524: `are labeled`


**Summary**

In general, I really like the idea presented in this paper and the overall execution of the experimental evaluation. The problem (data imbalance) is crucial in many real-world applications, and the proposed framework seems like a nice and lightweight solution. Nevertheless, I'd like the authors to address my concerns, especially in terms of another proof-read round and points W1 + W2.

I am willing to raise my score, after addressing my concerns.

---

### Meta-Review · Area_Chair_Lird · 2022-11-13

**Confidence:** 4
**Recommendation:** Accept

**Meta Review:**

This paper proposes TopoImb, a novel architecture that seeks to attend to and modulate topology-level imbalance in graph representation learning. Reviewers have reached a consensus that the problem setting is novel, the paper is well-motivated, and the experiments are extensive. The main weaknesses were regarding the clarity of the proposed method, questions on the experimental results, and the discussions with the related work. The authors have successfully stressed the concerns during the paper discussion phase.

Here are the critical points used for the paper's decision:

S1. The problem setting is new and important

S2. The experiments are extensive

W1. The clarity of the proposed methods can be improved

Overall, S1>S2>W1, therefore I recommend the acceptance of this paper.

---

### Decision · Program_Chairs · 2022-11-22

Accept (Poster)